# A Novel Combination of Bevacizumab with Chemotherapy Improves Therapeutic Effects for Advanced Biliary Tract Cancer: A Retrospective, Observational Study

**DOI:** 10.3390/cancers13153831

**Published:** 2021-07-29

**Authors:** Sung-Nan Pei, Chun-Kai Liao, Yaw-Sen Chen, Cheng-Hao Tseng, Chao-Ming Hung, Chong-Chi Chiu, Meng-Che Hsieh, Yu-Fen Tsai, Hsiu-Yun Liao, Wei-Ching Liu, Kun-Ming Rau

**Affiliations:** 1Department of Hematology-Oncology, E-Da Cancer Hospital, Kaohsiung 824, Taiwan; ed112124@edah.org.tw (S.-N.P.); ed111216@edah.org.tw (M.-C.H.); ed112116@edah.org.tw (Y.-F.T.); hsiuyun30@gmail.com (H.-Y.L.); a0920365248@gmail.com (W.-C.L.); 2School of Medicine, College of Medicine, I-Shou University, Kaohsiung 824, Taiwan; ed102489@edah.org.tw (Y.-S.C.); ed104817@edah.org.tw (C.-H.T.); ed100647@edah.org.tw (C.-M.H.); chiuchongchi@gmail.com (C.-C.C.); 3Division of Hematology and Oncology, Department of Internal Medicine, Chang Gung Memorial Hospital, Kaohsiung 833, Taiwan; Chunkai.liao@gmail.com; 4Division of General Surgery, Department of Surgery, E-Da Hospital, Kaohsiung 824, Taiwan; 5Division of Gastroenterology and Hepatology, E-Da Cancer Hospital, Kaohsiung 824, Taiwan; 6Department of General Surgery, E-Da Cancer Hospital, Kaohsiung 824, Taiwan; 7Kaohsiung Nan-Ping Cancer Prevention and Education Association, Kaohsiung 807, Taiwan

**Keywords:** advanced biliary tract cancer, bevacizumab, vascular normalization, gemcitabine, response rate

## Abstract

**Simple Summary:**

Systemic therapies for advanced biliarty tract cancers (BTC) are limited. The combination of gemcitabine with cisplatin (GC) has been the standard first-line treatment for advanced BTC from 2010 until now. In order to improve therapeutic effect, especially response rate, we added a novel schedule and dosage of bevacizumab to standard GC regimen. In our real world date, we found this regimen could increase the overall response rate to 50.0%, and side effects were managable. For patients with advanced BTC, especially whose tumors need rapid response to treatment, our regimen can provide an alternative choice.

**Abstract:**

Background: Biliary tract cancer (BTC) is a heterogenous collection of biliary tract cancer at different primary sites, and the prognosis of advanced BTC is dismal. Systemic chemotherapy with gemcitabine and cisplatin (GC) has been the reference regimen since 2010. How to improve therapeutic effects of GC regimen is an urgent mission at present. Methods: Bevacizumab with a reduced dosage and modified schedule (10 mg/Kg/triweekly, 1 day before GS at the first 2 cycles) was combined with standard GC for patients with advanced BTC. Tumor response was assessed using Response Evaluation Criteria in Solid Tumors version 1.1 every 2 months. Kaplan–Meier curves were estimated for time-to-treatment failure (TTF), progression-free survival (PFS) and overall survival (OS). Result: A total of thirty cases of advanced BTC accepted this treatment, and the overall response rate (ORR) was 50.0%, and the disease control rate was 80.0% for all patients. The median TTF was 5.8 months, the median PFS was 8.4 months, and the median OS was 13.6 months. Most responses were noted at the first evaluation. Adverse effects (AEs) were mostly tolerable. Conclusions: After modifying the schedule, adding bevacizumab to a traditional GC regimen could increase the ORR with a shorter time-to-response, a better PFS and OS than GC alone but without the addition of AE. This regimen can be applied to patients with advanced BTC, especially those who are with a big tumor burden and who need a rapid response.

## 1. Introduction

Biliary tract cancer (BTC) represents a collection of disease entities that comprise a heterogeneous group of adenocarcinomas including gallbladder cancer (GBC), cholangiocarcinoma of intrahepatic bile ducts (iCCA) and extrahepatic bile ducts (eCCA), perihilar (pCCA), and the ampulla of Vater (AVC) [1]. BTC accounts for approximately 3% of all adult gastrointestinal malignancies, representing the secondary most common hepatobiliary cancer following hepatocellular carcinoma (HCC) [2], and its incidence is increasing [3]. Another rare type of primary liver cancer is a mixture of HCC and iCCA, which has the worst overall survival (OS) [4]. Compared to Western countries, the incidence of BTC in Asia is extremely high and can be up to 71.3 per 100,000 in certain parts of Asia [5].

Each subtype of BTC implies a distinct epidemiology, clinical symptoms, molecular biology, prognosis, strategy for clinical management, and risk factors. These factors include fluke, primary sclerosing cholangitis, liver cirrhosis (LC), hepatitis C infection (HCV), hepatitis B infection (HBV), metabolic syndrome, and diabetes [6]. A history of cholecystolithiasis is the strongest risk factor for GBC [7]. A meta-analysis of several studies on the risk factors for iCCA showed the following associations: LC had a combined odds ratio (OR) of 22.92 (95% CI 18.24–28.79), HCV of 4.84 (2.41–9.71), and HBV of 5.10 (2.91–8.95) [8].

Complete surgical resection is the mainstay cure in early stages of BTC, but unfortunately, only 10 to 40% of patients are diagnosed with resectable disease [9]. Curative surgical resection with negative tumor margins can be achieved in less than 30% of patients [10]. The majority of patients with BTC present with unresectable disease and show either locally advanced or distant metastasis. Patients who undergo potentially curative surgery still experience a high rate of relapse [11]. Survival rates of five years are less than 5–10% for advanced BTC, and the median OS of patients with advanced disease is frequently less than one year [6].

From two parallel trials, the Advanced biliary tract cancer (ABC)-02 study (ClinicalTrials.gov number: NCT00262769) [12] and the biliary tract (BT) 22 study (ClinicalTrials.gov number NCT00380588) [13], compared doublet-chemotherapy (GC, gemcitabine 1000 mg/m^2^ and cisplatin 25 mg/m^2^ each on days 1 and 8 of a 21-day regimen) versus gemcitabine monotherapy (Gem, gemcitabine 1000 mg/m^2^ on days 1, 8 and 15 of a 28-day regimen). A meta-analysis of ABC-02 and BT 22 found that GC demonstrates a significant improvement in the progression free survival (PFS) (median PFS 8.8 vs. 6.7 months; hazard ratio (HR) = 0.64, 95% confidence interval (CI) 0.53–0.76, *p* < 0.001) and overall survival (OS, ABC-02 11.7 vs. 8.1 months; BT-22 11.2 vs. 7.7 months; HR = 0.65, 95% CI 0.54–0.78, *p* < 0.001) over the Gem. Response rate (RR), which ranged from 19.1%(BT-22) to 26.1%(ABC-02). The disease control rate (DCR) ranged from 68.3% (BT-22) to 81.4% (ABC-02). GC resulted in improved RR, PFS, and OS for advanced BTC, including iCCA, eCCA, and GBC, thus setting a reference regimen for patients with advanced BTC [14].

After these studies, there were several new regimens that combined either target therapies or immunotherapies to GC. The vascular endothelial growth factor (VEGF) is expressed in approximately 50% of iCCA [15]. Bevacizumab, an anti-VEGF antibody, exhibits an ability to normalize peritumoral vessels in preclinical metastatic models, leading to an enhanced delivery of cytotoxic agents to the tumor [16]. The combination of bevacizumab with chemotherapeutic agents has proven effective for treating breast, lung, and colon cancers. As the first line therapy for advanced BTC, a combination of bevacizumab with chemotherapy also showed promising results with an overall response rate (ORR) of 40% and a median OS of 12.7 months [17].

In an in vivo xenograft study, the administration of bevacizumab 1–3 days before chemotherapy in tumor-bearing mice resulted in a higher intratumoral chemotherapy penetration and tumor-growth inhibition compared to the concomitant administration of the two drugs [18]. A phase II study using bevacizumab combined with etoposide and cisplatin for patients with brain metastases caused by breast cancer, bevacizumab was given 1 day before chemotherapy during the first two treatment cycles. A total of twenty-seven patients (77.1%) achieved an ORR for intracranial tumors. The median PFS and OS were 7.3 months (95% CI, 6.5–8.1) and 10.5 months (95% CI, 7.8–13.2), respectively [19].

In order to enhance the response rate, we added bevacizumab to a standard GC regimen for advanced BTC. During the first two cycles, bevacizumab was given 1 day before chemotherapy. In this retrospective study, we collected and analyzed the therapeutic effects in our real-world practice.

## 2. Materials and Methods

### 2.1. Study Population

Clinical data of patients who received bevacizumab, gemcitabine, and cisplatin (A-GC) between Aug. 2018 and Mar. 2021 at the E-Da Cancer Hospital were retrospectively collected and reviewed. The study design was approved by Institutional Review Board (EMRP-110-063) of E-Da Cancer Hospital.

Inclusion criteria were pathologically confirmed BTC that was either recurrent after operation, de novo locally advanced, or de novo stage IV. Patients had to be treated with at least 1 cycle of an A-GC regimen. Chart review and data collection included medical history, tumor characteristics, clinical parameters (e.g., primary site, site of metastases), blood test (complete blood cell count, biochemistry and tumor markers), treatment events (e.g., number of therapeutic cycles, start/end dates, and rationale for discontinuation), clinical response, use of supportive care medications (e.g., granulocytic colony stimulating factor), dose adjustments, and adverse events. The reported results were based on the effectiveness analysis of the data collected by June 2021.

### 2.2. Treatment

Patients received bevacizumab (10 mg/kg; day 1), gemcitabine (1000 mg/m^2^/day; days 2, 8), and cisplatin (25 mg/m^2^; day 2, 8). Treatment was based on a 21-day cycle. After 2 cycles, gemcitabine and cisplatin were changed to days 1 and 8. However, it was discontinued if a patient exhibited disease progression, showed intolerable toxicities, or died.

### 2.3. Outcome Measures

The primary outcome measure was ORR, which was defined as the proportion of patients who achieved a complete response (CR), a partial response (PR), and DCR, which was defined as the proportion of patients who achieved CR, PR, and stable disease (SD) as the best response. Tumor response was assessed using the Response Evaluation Criteria in Solid Tumors version 1.1, with computed tomography scans or magnetic resonance imaging at baseline and every 2 months. Bone scans and chest X-rays were used as adjuvant evaluation tools. Thus, the percentages of patients with advanced BTC who achieved CR, PR, and SD during A-GC treatment were recorded. The safety of the A-GC was evaluated by the number of patients with adverse events (AEs) and the severity of the AEs, which were assessed using the Common Terminology Criteria for Adverse Events (CTCAE), version 4.0. This included all events that were not present before the initial administration of A-GC, pre-existing events that became more intense or more frequent, and events that were present upon initial A-GC administration but became more severe following administration.

### 2.4. Statistical Analysis

The characteristics of the patients and tumors, the treatment duration, the tumor response, and other categorical variables were summarized as number and percentage and age as median (range). The time to treatment failure (TTF) was defined as the period from the first dose of A-GC to cancellation for any reason including death, disease worsening, treatment toxicity, patient request, or was censored at the date of the last follow-up for surviving patients remaining on treatment. PFS was defined as the time from treatment initiation to either progression or death.

The overall survival time was defined as the period from the first dose of A-GC to the date of patient death, loss of follow-up, or the date of the last follow-up for surviving patients. Time-to-event endpoints were summarized using the Kaplan–Meier method.

## 3. Results

### 3.1. Patient Characteristics

From August 2018 to March 2021, there were a total 30 patients who had been diagnosed with advanced BTC who received this regimen. A summary of the patient characteristics is shown in Table 1.

The median age at treatment was 58.0-year-old. Most patients had good performance status at the time of treatment. Intrahepatic cholangiocarcinoma was predominant. Because Taiwan is an epidemiologic area of hepatitis, there were 13 (43.3%) cases involving hepatitis. Only eight (26.7%) cases had accepted a curative operation before, and most patients were diagnosed at locally advanced or distant metastasis stages (25, 83.3%). Liver and lymph nodes (LNs) were the most common sites of metastasis.

Although the liver was the most common site of metastasis, the liver functions in most of the cases were still within normal limits, as presented in Table 2. Compared to the carcinoembryonic antigen (CEA), more cases had the abnormal carbohydrate antigen 19-9 (CA 19-9).

### 3.2. Treatment Efficacy

The ORR was 50.0%, and the DCR was 80.0% for all patients. No patient achieved CR (Table 3). The response evaluation of two ampulla of Vater patients was one PR and one SD. The median TTF was 5.8 months (Figure 1A), the median PFS was 8.4 months (Figure 1B), and the median OS was 13.6 months (Figure 1C).

### 3.3. Safety

Adverse effects were collected from chart review (Table 4). In general, this regimen did not have severe bone morrow suppression. About 25% cases experienced either neutropenia or thrombocytopenia, but none of them needed the granulocyte stimulating factor. Because bevacizumab is not reimbursed by health system here, patients had to pay for this treatment themselves, and several cases chose to stop this regimen after good control of their tumors was achieved. The median of the A-GC cycles was 6. The most common reason for discontinuation was PD followed by AE such as thrombocytopenia or paresthesia and economic problems (Table 5).

After the discontinuation of A-GC, patients with good performance status were able to accept the secondary treatment which included oxaliplatin (mFOLFOX6), irinotecan (FOLFIRIL), S-1, and other drugs (Table 6).

## 4. Discussion

Although there have been many improvements in the treatment choices and survival rates of different cancers, BTC remains one of the most dismal tumors with very limited therapeutic options. Adjuvant therapies for BTC, including chemotherapy, radiotherapy, or the combination of these two are common, though the benefit of post-operative treatment is somewhat unclear due to conflicting results of randomized trials. Recently, two meta-analyses confirmed the benefit of adjuvant chemotherapy and chemoradiotherapy, but for patients with positive regional LNs or surgical margin, the DFS and OS are still poor [20,21]. After the ABC-02 and BT 22 reports in 2010, GC became the reference regimen of the first-line systemic treatment of advanced BTC patients until now.

Compared to gemcitabine alone, GC not only improved ORR but also improved PFS and OS. In order to further improve therapeutic effects, coming GC with target therapy, immunotherapy, or different chemotherapy regimens have been developed, and several studies are still ongoing [22].

In our treatment, we added bevacizumab to a standard GC regimen. The dosage and schedule of bevacizumab were different from regular practices. The first difference was the dosage of bevacizumab. Bevacizumab has different dosages for different cancers. For example, in colon cancer, the suggested dosage of bevacizumab is 2.5 mg/Kg/week [23]. In other cancers such as lung, breast, and hepatoma cancers the recommended dosage of bevacizumab is 5 mg/Kg/week [24]. Because bevacizumab is not reimbursed by National Health Insurance in Taiwan, patients have to buy this drug by themselves or pay for it through private insurance if allowed, so we modified the dosage of bevacizumab to 10 mg/Kg/triweekly, which should not be too low to be ineffective, and the price was acceptable for most patients. Although we decreased the dosage, we still could see the addictive effect of bevacizumab without more accumulated toxicities.

The second difference was the bevacizumab schedule. From our previous experience with breast cancer, bevacizumab prescribed one day before chemotherapy could lead to high RR, which we believed was from the effect of the normalization of tumor vessels [19]. Normalized vessels can have better drug delivery, which leads to a higher concertation of the treatment drugs in the tumors, which might be able to be translated into higher RR and a shorter time to response. In our treatment, we gave bevacizumab 1 day before GC during the first two cycles. Because of the long half-life of bevacizumab, the level of bevacizumab in the blood should be stable after two treatment cycles [25]. For the convenience, we combined bevacizumab with GC on day 1from cycle 3 onwards.

Compared to ABC-02 and BT 22, both trials only used chemotherapy, and our A-GC had a higher ORR (50%) and a similar DCR (80%). Most responses were noted at the first evaluation. In our treatment, the first response evaluation was at around week 9, before the start of cycle 4. The first evaluation was at week 12 in ABC-02 and at week 6 in BT-22. Adding bevacizumab to GC, we see not only a better ORR but also a faster response. For patients who have a big tumor burden or impaired liver function from the mass effect, shorter time-to-response (TTR) is important. Although our median TTF was only 5.8 months, some patients stopped treatment due to AEs, economic problems, and other causes such as traffic problems. All of these conditions would happen in real-world. The median PFS in our study was 8.4 months, equal to ABC-02 and better than BT-22. Most importantly, the OS in our study was longer than 1 year, continuing up to 13.6 months.

Until now, there are only a few studies combining bevacizumab with chemotherapy as the first line of treatment for advanced CCA. For example, Lyer et al. reported a phase II study combining bevacizumab with gemcitabine and capecitabine for advanced CCA. The RR was 24%, the CBR was 72%, the median PFS 8.1 months, and the median OS 10.2 months. Adding bevacizumab did not improve outcome in an unselected group of patients with advanced BTC [26].

Other target therapies, such as erlotinib, cetuximab, panitumumab, sorafenib, cediranib, and trametinib had been conducted in clinical trials, but none were shown to be effective for advanced BTC [27].

Since our GC dose and schedule was the same as those from the reference trials, the reason why our treatment could have a better ORR not only be from adding bevacizumab but also from different the schedule during the first two cycles. Although almost all of our cases were iCCA, no differences in the rate of response between the GBC and cholangiocarcinoma subgroups were reported in ABC-02 [12].

Different chemotherapy such as oxaliplatin, S-1, and Nab-paclitaxel with doublet combinations have been reported as the first line of systemic therapy. Treatment effects were not superior to GC. Triplet combinations such as oxaliplatin, irinotecan, and S-1; gemcitabine, cisplatin, and Nab-paclitaxel; or gemcitabine, cisplatin, and S-1 could produce a higher RR and OS, but the toxicities, especially in the myelosuppression and the gastrointestinal tract, were higher than those seen in GC [27,28,29].

Since BTC is a kind of carcinoma, drugs that inhibit the epidermal growth factor receptor pathways such as erlotinib, cetuximab or panitumumab had also been used in combination with chemotherapy. None of them provided a survival benefit compared to GC alone [29], which might indicate the selection of predictive biomarkers [30].

BTC was known to be an immune desert with low tumor mutation burden and programmed cell death protein 1(PD-L1) expression [31,32]. At present, the clinical data on immunotherapy in BTC are limited. Trials of monotherapy all focused on a later line, with limited RR and survival benefits [33,34]. A recently published expert opinion also suggested that immunotherapy should not be considered as a preferred systemic treatment in BTC patients with microsatellite stable disease outside of clinical trials. However, they also pointed out that immunotherapy-based combinations and highlight pivotal studies will likely influence the future development of relevant concepts in BTC [35]. These trials include the phase III, double-blind TOPAZ-1 study, which randomly allocates treatment-naïve patients to GC plus durvalumab versus GC in combination with placebo, and the KEYNOTE-966 study, which is currently evaluating the role of pembrolizumab combined with GC versus GC plus placebo (Table 7) [36]. In the coming years, the first-line therapies for advanced BTC should have great improvements.

The dilemma of systemic therapies for advanced BTC is not only limited to first line treatments but is also a more difficult situation at the later lines. Although GC has created the basis as first line treatment choice, after tumors progress, there are no standard secondary or later line therapies. Fluoropyrimidine-based combination therapy with either oxaliplatin (FOLFOX) or irinotecan (FOLFIRI) showed a median PFS of 1.6–3.9 months and a median OS of 4.4–8.4 months [37,38]. The most recently published results from ABC-06, which was an open-label, phase III trial comparing 12 cycles of mFOLFOX plus active symptom control (ASC) with ASC alone in the second line setting, the RR was 5% and the median PFS was 4.0 months in the ASC plus FOLFOX group. The median OS was significantly longer in the ASC plus FOLFOX group than in the ASC alone group (6.2 vs. 5.3 months, HR 0.69 *p* = 0.031) [39].

Through the improvement and prevalence of next generation sequences (NGS), many driven genes or mutations are being identified, helping to explain the underlying mechanism of the pathogenesis of BTC and to develop new therapies. After analysis using NGS, we know that BTC at different locations are not only clinically heterogeneous but are also genetically heterogeneous [40]. For example, isocitrate dehydrogenase (IDH) and FGFR aberrations tend to cluster in iCCA, whereas HER2 (ERBB2) aberrations are more frequent in eCCA and GBC [41]. Lowery et al. found that genetic alterations with potential therapeutic implications were identified in 47% of BTC patients, resulting in biomarker-directed therapy being possible [42]. In clinical trials, targeted therapies against druggable molecular alterations, such as fibroblast growth factor receptor (FGFR)-2 fusions and IDH-1 mutations, showed promising results [22,43]. After 10 years of chemotherapy being the only standard option for patients with advanced BTC, the first FGFR-2 fusion or rearrangement inhibitor pemigatinib gained regulatory approval for previously treated, locally advanced or metastatic BTC in April 2020 [44]. The guidelines of the European Society for Medical Oncology already recommend the routine use of NGS on tumor samples in advanced CCA [45]. In the future, molecular profiling studies will better delineate the genetic landscape of each BTC subtype, highlighting distinct patterns of mutations and therapies in specific anatomic subtypes.

The limitations of our study result from its retrospective nature and no formal schedule, especially in regard to the number of cycles. However, to our knowledge, this retrospective study reported the largest number of advanced BTC patients receiving A-GC in a real-world setting. Although we found that A-GC can be an effective treatment for advanced BTC, a confirmation study is still needed prior making a conclusion regarding activity of the combination. After modifying the schedule, the addition of bevacizumab to GC can shorten TTR and increase ORR without adding toxicities. Such benefits can be achieved even with decreased dosage of bevacizumab. For patients who need rapid tumor response or who are of a limited medical budget, A-GC can be a reasonable choice.

## 5. Conclusions

Effective treatments for advanced BTC are still lacking, and many patients are diagnosed with an extensive disease burden. After modifying of the schedule, the addition of bevacizumab to GC shortened TTR and increased ORR without adding toxicities. Such benefits can be achieved even with a decreased dosage of bevacizumab. For patients who need rapid tumor response or who are of a limited medical budget, A-GC can be a reasonable and cost-effective choice.

## Figures and Tables

**Figure 1 cancers-13-03831-f001:**
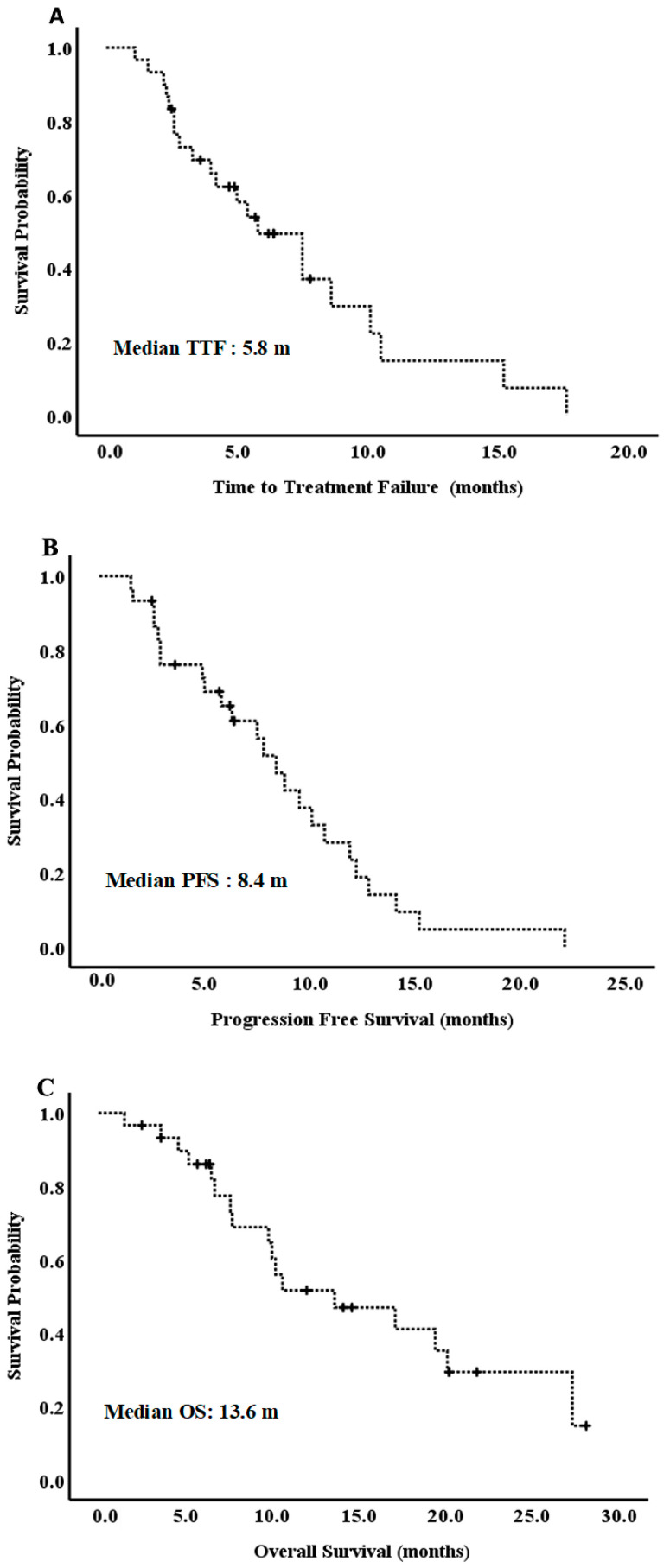
(**A**). Time-to-treatment failure (TTF); (**B**). progression free survival (PFS); and (**C**), overall survival (OS).

**Table 1 cancers-13-03831-t001:** Clinical characteristics of patients at baseline.

Characteristics	N (%)
Age at treatment (median)	58 years old (39–70)
Sex	
Male	17 (56.7)
Female	13 (43.3)
ECOG performance status	
0	4 (13.3)
1	20 (66.7)
2	6 (20)
Primary site	
Intrahepatic	28 (93.3)
Ampulla of Vater	2 (6.7)
Initial stage	
I	2 (6.7)
II	3 (10.0)
III	6 (20.0)
IV	19 (63.3)
Combined liver disease	
HBV	10 (33.3)
HCV	2 (6.7)
HBV + HCV	1 (3.3)
Alcohol	2 (6.7)
None	15 (50.0)
Liver cirrhosis	
Yes	2 (6.7)
No	28 (93.3)
Curative operation before	
Yes	8 (26.7)
No	22 (73.3)
Adjuvant chemotherapy for curatively operated cases	
Yes	4 (13.3)
No	4 (13.3)
Disease status at treatment	
Locally advanced only	3 (10.0)
With distant metastasis	
Liver	15 (50.0)
Regional LN	12 (40.0)
Distant LN	9 (30.0)
Bone	3 (10.0)
Peritoneum	1 (3.3)
Lung	7 (23.3)
Others	1 (3.3)

ECOG: Eastern Cooperative Oncology Group, HBV: hepatitis B virus, HCV: hepatitis C virus, LN: lymph node.

**Table 2 cancers-13-03831-t002:** Biologic characteristics at baseline.

Biologic Characteristics	Median (Range)
Albumin, g/L	3.9 (2.6–4.7)
AST, UI/L	30 (17–94)
ALT, UI/L	21 (8–87)
Total bilirubin mg/dl	0.82 (0.26–4.04)
Alkaline phosphatases, UI/L	350 (65–1878)
CEA	3.79 (0.75–5170)
CA 19-9	78.41 (2–150960)
N/L ratio	4.2 (0.9–14.7)

AST: Aspartate transaminase, ALT: alanine aminotransferase, CEA: carcinoembryonic antigen, CA19-9: carbohydrate antigen 19-9, N/L: ratio neutrophil to lymphocyte ratio.

**Table 3 cancers-13-03831-t003:** Best therapeutic response.

Best Response Evaluated	N (%)
Partial response	15 (50.0)
Stable disease	9 (30.0)
Progressive disease	4 (13.3)
Not evaluable	2 (6.7)

**Table 4 cancers-13-03831-t004:** Most severe toxicities from treatment.

Toxicities	N (%)
Neutropenia	
Gr 1, 2	17 (56.7)
Gr 3, 4	7 (23.3)
Anemia	
Gr 1, 2	23 (76.7)
Gr 3, 4	3 (10.0)
Thrombocytopenia	
Gr 1, 2	13 (43.3)
Gr 3, 4	8 (26.7)
Hepatobiliary disorder	
Gr 1, 2	13 (43.3)
Gr 3, 4	3 (10.0)
Peripheral neuropathy	
Gr 1, 2	10 (33.3)
Gr 3, 4	2 (6.7)
Hypertension	
Gr 1, 2	15 (50.0)
Gr 3, 4	0
Constipation	
Gr 1, 2	9 (30.0)
Gr 3, 4	5 (16.7)
Skin rash	
Gr 1, 2	7 (23.3)
Gr 3, 4	1 (3.3)
Alopecia	
Gr 0, 1	10 (33.3)
Gr 2	0
ausea	
Gr 1, 2	15 (50.0)
Gr 3, 4	2 (6.7)
Vomiting	
Gr 1, 2	10 (33.3)
Gr 3, 4	1 (3.3)

Gr: grade.

**Table 5 cancers-13-03831-t005:** Reasons for discontinuation.

Reason	N (%)
PD	11 (36.7)
AE	6 (20.0)
Economic	3 (10.0)
Others	5 (16.7)

PD: progressive disease, AE: adverse effect.

**Table 6 cancers-13-03831-t006:** Second line therapy after disease progression.

Regimen	N (%)
FOLFOX	5
FOLFIRI	2
S-1	5
Supportive care	6
Others	7

FOLFOX: oxaliplatin-based regimen, FOLFIRI: irinotecan-based regimen.

**Table 7 cancers-13-03831-t007:** Results of published/ongoing first line treatments for advanced biliary tract cancer.

Regimen	Phase	Patient Number	Gr 3, 4 Neutropenia (%)	ORR/DCR (%)	Median PFS (m)	Median OS (m)
Modified Bevacizumab plus GC (Rau et al.)	Retrospective	30	23.3	50.0/80.0	8.4	13.6
GC (ABC-02) [12]	III	198	25.3	26.1/81.4	8.0	11.7
Bevacizumab plus gemcitabine and capecitabine [26]	II	50	36	24/72	8.1	10.2
Oxaliplatin, Irinotecan, and S-1 [28]	II	32	32	50/88	6.8	12.5
Anlotinib plus sintilimab plus GC [36]	II	80	Pending	Pending	Pending	Pending
Nab-paclitaxel, gemcitabine, plus cisplatin [29]	II	60	33	45/84	11.8	19.2
Pembrolizumab plus GC (KEYNOTE 966) [36]	III	788	Pending	Pending	Pending	Pending
Durvalumab plus GC (TOPAZ-1) [36]	III	757	Pending	Pending	Pending	Pending

GC: gemcitabine plus cisplatin, Gr: grade, ORR: overall response rate, DCR: disease control rate, PFS: progression free survival, OS: overall survival.

## Data Availability

The data presented in this study are available upon request from the corresponding author. The data are not publicly available due to regulations of the institution.

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
