# Peer review of "A Novel Combination of Bevacizumab with Chemotherapy Improves Therapeutic Effects for Advanced Biliary Tract Cancer: A Retrospective, Observational Study"

_cancers, 2021, doi:10.3390/cancers13153831_

Round 1

Reviewer 1 Report

The study is a single arm retrospective study for patients with advanced cholangiocarcinoma received bevacizumab with chemotherapy. The study is well explained and discussed. The major issue is the lack of comparison group in analysis. So the authors claim "improves therapeutic effects" can not be justified in their study. The authors are encouraged to compare this novel combination with other regimens or best supportive care to provide more scientific soundness. 

Author Response

Author response: Thanks for reviewer’s wonderful suggestion and comment. Of course, it is reasonable to conduct a clinical trial with a balanced control group, but unfortunately there was no pharmaceutical company would support us to conduct a phase II study because a phase II study could not get any approval, especially there were already several bevacizumab-combination trials before, and the patent of bevacizumab was already expired. In our manuscript, we tried to compare our data with other first-line therapies, not only chemotherapy, but also combinations including target therapy and immunotherapy. By your kindly suggestion, we added table 7 which included several import trials, these trials are either already published or still ongoing. We also added a statement “Although we found A-GC can be an effective treatment for advanced BTC, confirmation study is still needed prior making conclusion of the activity of the combination.” into discussion. In this manuscript, we focused on the first line, systemic therapies, so we did not mention too many about FGFR inhibitors and secondary line therapies. Although this is a retrospective study, it is a real world data. After modifying the schedule of bevacizumab, we did see higher response rate, such finding made us very surprise and want to share with other physicians in the world.

Reviewer 2 Report

I would like to congratulate the authors of the paper for their efforts.

However in my opinion, changes have to taken place prior accepting the article:

1) I would prefer from you to use the word "biliary tract cancer" rather than "cholangiocarcinoma" as cholangiocarcinoma does not include ampullary carcinoma and gallbladder carcinoma.

2) I am concerned regarding the type of research. You have used Bevacizumab which has not indication for biliary tract cancer. How have you used it for our patients? Maybe you need to explain that better (hw you have ginven to the patients bevacizumab) or you might want to state that you did a phase II study if that was the case.

3)Maybe you would like to exclude the 2 patients with ampullary carcinoma as the other 28 had intrahepatic cholangiocarcinoma or at least to say how the ampullary cancer patients responded to treatment.

4) In the discussion you should not be so definite that the combination is so effective. I would rather prefer from you to state that "confirmation study is needed prior making conclusion of the activity of the combination. 

5) In line 23 you should write " GC" rather than "GS"

Thank you

J Sgouros

Author Response

We deeply appreciate these kindly and creative feedback and comment from distinguished reviewer. Below are our response point-by-point. 

1) I would prefer from you to use the word "biliary tract cancer" rather than "cholangiocarcinoma" as cholangiocarcinoma does not include ampullary carcinoma and gallbladder carcinoma.

Author response: Thanks for this kindly comment, I already changed cholangiocarcinoma (CCA) to biliary tract cancer (BTC) in the manuscript title and text.

2) I am concerned regarding the type of research. You have used Bevacizumab which has not indication for biliary tract cancer. How have you used it for our patients? Maybe you need to explain that better (hw you have ginven to the patients bevacizumab) or you might want to state that you did a phase II study if that was the case.

Author response: Although bevacizumab was not got approved by any FDA, from several trials, adding bevacizumab still provided borderline effects. Because bevacizumab was known to have normalization effect on tumor vessel, and our novel schedule had been proved which could enhance therapeutic effect of chemotherapy on breast cancer, so I tried several cases at the first beginning, I did see dramatic effects. Unfortunately, no pharmaceutical company would support us to conduct a phase II study because a phase II study could not get any approval, especially there were already several bevacizumab-combination trials before and the patent of bevacizumab was already expired. Patients at Taiwan had to pay this drug by themselves. Luckly, many of them could get reimbursement from insurance companies by local practice. Our data was quite good, so I sincerely hope we can provide another choice for patients who can not offer novel agents such as FGFR inhibitor or immune check point inhibitor.

3) Maybe you would like to exclude the 2 patients with ampullary carcinoma as the other 28 had intrahepatic cholangiocarcinoma or at least to say how the ampullary cancer patients responded to treatment.

Author response: The best response of these 2 cases were PR and SD. As your kindly suggestion, I already added the statement into the section of result.

4) In the discussion you should not be so definite that the combination is so effective. I would rather prefer from you to state that "confirmation study is needed prior making conclusion of the activity of the combination. 

Author response: Thanks for the suggestion, I already add this description into the section of discussion.

5) In line 23 you should write " GC" rather than "GS"

Author response: Many thanks, I already made the correction.

Round 2

Reviewer 1 Report

Thank you for the revision.

Author Response

Author response: Thanks for your kindly suggestion and feedback.

Reviewer 2 Report

I would like to thank the authors for the changes they had made to their manuscript.

However I am still concerned about Bevacizumab. I would prefer prior making the paper accepted for publication, the authors to inform the readers the dose of Bevacizumab 10mg/Kg was chosen. Why they have used 10 and not 15mg/kg?

Also please in line 167 delete the phrase " The patient of GBC".

Kind regards

Author Response

I would like to thank the authors for the changes they had made to their manuscript.

  • However I am still concerned about Bevacizumab. I would prefer prior making the paper accepted for publication, the authors to inform the readers the dose of Bevacizumab 10mg/Kg was chosen. Why they have used 10 and not 15mg/kg?

Author response: Thanks for this critical question. As we know, the suggested dosage of bevacizumab varies between different cancers. Although in most cancers, the suggested dosage of bevacizumab is 5mg/Kg/week, in colon cancer, it is 2.5mg/Kg/week. Most interesting, in AVAiL trial for non-small cell lung cancer, the therapeutic effects of low dose bevacizumab (7.5mg/Kg/triweekly) were not inferior to high dose bevacizumab (15mg/Kg/triweekly). It means effects of bevacizumab might not necessarily correlate with dosage. The reasons why we choose the dosage of 10mg/Kg/triweekly are:

  1. There is no standard dosage of bevacizumab for BTC, but the dosage is near the recommended dosage for most cancers. It is not too low to be accepted. At least, we hope this higher dosage might help us to prevent too many criticisms.
  2. Because patients have to pay bevacizumab, under this dosage, the price of bevacizumab can be accepted by most patients if they have private insurance. It is a real-world situation.
  3. Since we did see good effect under this dosage, and we wanted patients could use bevacizumab as long as possible even after we changed to other regimen, so we chose this dosage to delay the appearance of bevacizumab-related side effects.

We added” which should not be too low to be ineffective and the price was acceptable by most patients.” at line 284 of discussion as your kindly suggestion.

  • Also please in line 167 delete the phrase " The patient of GBC".

Author response: Thanks for this kindly comment, but I am sorry that I could not see this phrase in my manuscript.